# UAV Photogrammetry in Intertidal Mudflats: Accuracy, Efficiency, and Potential for Integration with Satellite Imagery

Chunpeng Chen [1,2], Bo Tian [1], Wenting Wu [3], Yuanqiang Duan [1], Yunxuan Zhou [1] and Ce Zhang [2,4,*]

1    State Key Laboratory of Estuarine and Coastal Research, East China Normal University, Shanghai 200241, China
2    Lancaster Environment Centre, Lancaster University, Lancaster LA1 4YQ, UK
3    Key Laboratory of Spatial Data Mining and Information Sharing of Ministry of Education, National & Local Joint Engineering Research Center of Satellite Geospatial Information Technology, Fuzhou University, Fuzhou 350108, China
4    UK Centre for Ecology & Hydrology, Library Avenue, Lancaster LA1 4AP, UK
*    Correspondence: c.zhang9@lancaster.ac.uk

**Abstract:** The rapid, up-to-date, cost-effective acquisition and tracking of intertidal topography are the fundamental basis for timely, high-priority protection and restoration of the intertidal zone. The low cost, ease of use, and flexible UAV-based photogrammetry have revolutionized the monitoring of intertidal zones. However, the capability of the RTK-assisted UAV photogrammetry without ground control points, the impact of flight configuration difference, the presence of surface water in low-lying intertidal areas on the photogrammetric accuracy, and the potential of UAV/satellite Synergy remain unknown. In this paper, we used an RTK-assisted UAV to assess the impact of the above-mentioned considerations quantitatively on photogrammetric results in the context of annual monitoring of the Chongming Dongtan Nature Reserve, China based on an optimal flight combination. The results suggested that (1) RTK-assisted UAVs can obtain high-accuracy topographic data with a vertical RMSE of 3.1 cm, without the need for ground control points. (2) The effect of flight altitude on topographic accuracy was most significant and also nonlinear. (3) The elevation obtained by UAV photogrammetry was overestimated by approximately 2.4 cm in the low-lying water-bearing regions. (4) The integration of UAV and satellite observations can increase the accuracy of satellite-based waterline methods by 51%. These quantitative results not only provide scientific insights and guidelines for the balance between accuracy and efficiency in utilizing UAV-based intertidal monitoring, but also demonstrate the great potential of combined UAV and satellite observations in identifying coastal erosion hotspots. This establishes high-priority protection mechanisms and promotes coastal restoration.

**Keywords:** intertidal topography; accuracy; flight optimization; UAV/satellite synergy

## 1. Introduction

Intertidal mudflats, located in the sensitive transitional zone between marine and terrestrial systems, are of great social, economic, and environmental importance at a global scale. They support extensive habitats for migratory birds, provide potential land resources for regional development, and buffer against natural disasters from oceans [1–3]. With the continuous initiation, transport, and deposition of sediments under the joint effects of tides, currents, and waves, mudflat geomorphology is changing constantly [4,5]. The mudflat geomorphology can also be reshaped by high-energy storms (e.g., typhoons) within a short time frame [6,7]. In turn, topographic changes among mudflats can affect hydrodynamics and sediment distribution [8]. High-resolution mudflat topography with a high level of vertical accuracy is beneficial to understanding such coastal geomorphological processes and identifying coastal erosion hotspots.

Accurate mudflat topographic data is also fundamental to morphodynamics simulation and tidal wetlands restoration. The creation and/or restoration of salt marshes, also known as nature-based solutions (NbS), is becoming increasingly economically and ecologically viable for reducing the risk of coastal hazards [9,10]. Small topographic differences (i.e., micro-topography) often determine the extent, duration, and frequency of tidal inundation, thereby affecting benthos distribution and salt marsh vegetation colonization on bare tidal flats [11,12]. Mudflats accumulate sediment constantly and experience seaward progradation under conditions of adequate sediment supply and they can provide more accommodation in terms of space for seaward expansion of salt marsh pioneer vegetation. In contrast, the erosion of mudflats brings uncertainty as to whether salt marshes can continue to survive in the context of sea level rise [13–15]. Van Regteren et al. [16] revealed that rapid changes in mudflat elevation may hinder the germination success of salt marsh vegetation through the burial of freshly sprouted seedlings. Thus, up-to-date topographic data for mudflats and the quantification of mudflat morphological changes are both critical to better understanding the geomorphological, hydrological, ecological, and hydrodynamic systems within intertidal zones.

Traditional field surveys for collecting intertidal mudflat topographic data, however, are often time-consuming and labor-intensive, spatially, and temporally constrained, with significant costs, and occasionally even impossible due to the poor accessibility and short exposure during tidal cycles. To overcome these limitations, a variety of remotely sensed techniques have been developed, including satellite-based waterline methods, video-based monitoring, terrestrial LiDAR (TLS), airborne LiDAR (ALS), and the emerging use of unmanned aerial vehicles (UAVs) for the purposes of structure-from-motion (SfM) photogrammetry (Table 1). The temporal-spatial resolution and accuracy of topographic data derived from these methods vary from daily to quarterly, and from centimeters to tens of meters. Particularly, TLS and ALS have been widely used in quantifying intertidal morphological changes thanks to their centimeter-level accuracy and fine spatial resolution. For example, Xie et al. [6] accurately tracked the erosion and deposition changes of intertidal mudflats at the timescale of a typhoon event using TLS. However, LiDAR data is hard to collect in areas with residual water or high water content on the mudflats at low tide, thereby, repeated scans are often required [17,18]. The time and costs involved in data collection result in a compromise in terms of frequency and spatial extent of observations. Instead, UAV-based SfM photogrammetry offers unique advantages in terms of cost, efficiency, flexibility, and data quality. It is, therefore, regarded as a revolutionary technology for topographic monitoring for hardly accessible coastal environments [19,20].

**Table 1.** Comparison of different topographic mapping methods in intertidal mudflats. V—vertical; H—horizontal.

| Technical Method | Spatial Resolution | Data Accuracy | Spatial Coverage | Repeatability | Limitations | Case References |
|---|---|---|---|---|---|---|
| Satellite-based waterline | ~30 m, depending on intervals of the waterline. | V: ~0.5 m; H: ~30 m | Large scale | Quarterly | Low accuracy; coarse temporal-spatial resolution; rely on good satellite observation. | [21–23] |
| Video-based monitoring | ~5 m, depending on intervals of the waterline. | V: ~0.5 m; H: ~5 m | ~1 km$^2$ of each camera | Daily | Low accuracy; cameras need to be installed at a high field of view. | [24–26] |
| Terrestrial LiDAR | ~0.5 m, depending on the sensor parameter. | V: ~4 cm; H: ~4 cm | ~1 km$^2$ of each station | Flexible | Costly; difficult to install in a muddy environment; few points are collected from sites where residual standing water remains. | [6,27] |
| Airborne LiDAR | ~0.5 m, depending on the sensor parameter. | V: ~13 cm; H: ~10 cm | Large scale | Flexible | Costly; few points are collected from sites where residual standing water remains. | [28–30] |
| UAV-based structure from motion | ~3 cm, depending on flight altitude and sensor parameter. | V: ~4 cm; H: ~3 cm | ~0.5 km$^2$ of each battery | Flexible | Data acquisition cannot be performed on rainy days. | [19,31,32] |

Despite the great potential of UAV-based SfM photogrammetry in monitoring intertidal topographic dynamics, there are some fundamental issues regarding the resultant topographic accuracy that have not been fully studied. First, SfM reconstructs the three-dimensional structure of a scene or object from a series of overlapping images acquired from different perspectives [33,34]. The selection of different flight parameters (e.g., flight pattern, altitude, and overlap) not only affects the efficiency of the aerial survey but also could lead to significant differences in the accuracy of the photogrammetric results. For example, Brunier et al. [35] quantified beach morphological changes based on images taken by a UAV at an altitude of 280 m, with an 85% frontal overlap and a 50% side overlap, with a vertical accuracy of about 10 cm; Chen et al. [31] investigated the morphological characteristics of tidal channels using UAV images acquired at 100-m altitude, with an 80% frontal overlap and a 70% side overlap, with a vertical accuracy of approximately 5.7 cm; and Kalacska et al. [19] constructed digital surface models for three salt marshes with a vertical accuracy of approximately 2.7 cm based on UAV images acquired at 30-m altitude, a 90% frontal overlap, and an 80% side overlap. However, the systematic evaluation of the effect of the difference in flight parameters on photogrammetric results was rarely mentioned in existing literatures, particularly in geographically distinctive mudflats. Second, in the case of low-accuracy UAV position and orientation system (POS) data, UAV-based photogrammetry is characterized by massive images, short baselines, irregular overlap, and considerable distortion. A large number of ground control points (GCPs) is needed to increase the accuracy of bundle adjustment [36]. However, the establishment of GCPs in muddy intertidal environments is impractical, and the number and spatial distribution of GCPs influence the accuracy significantly [37,38]. Recently, direct georeferencing of the UAV images with real-time kinematics (RTK) or post-processing kinematics (PPK), has the potential to accomplish UAV-based photogrammetry without GCPs by providing accurate and directly georeferenced surveys [39,40]. Although the usual flight configuration with nadir imaging may produce results with significant systematic elevation error [41], such error can be mitigated by using a small number of GCPs or by adding oblique imagery to the SfM workflow [42,43]. Yet, this RTK-assisted UAV-based photogrammetry is not well-researched in mapping estuarine intertidal topography. Third, in contrast with dry land surfaces, intertidal mudflats often present high water content under the influence of periodic tides, with residual water remaining on the surface. These water-bearing areas are generally considered to be non-Lambert and are highly anisotropic. As a result, variations in the angle of the images captured by the UAV could potentially lead to key points in these areas not being detected, thereby resulting in a failed estimation of elevation or presenting with huge uncertainty [44]. However, the tidal water remaining on the surface of the mudflats is shallow and turbid, and it is still unknown whether this would affect photogrammetric elevation estimates. In addition, the spatial extent of UAV observations is limited, and the ability to accurately construct large-scale intertidal topography in combination with satellite observations has rarely been investigated.

The limited exposure time for the intertidal zone makes the above issues very relevant to the accuracy and efficiency of UAV-based SfM photogrammetry. Considering the abovementioned issues, this study aims to quantify the difference in terms of topographic accuracy using RTK-assisted UAV-based SfM photogrammetry based on various photogrammetric scenarios, thereby facilitating the precise and efficient application of UAVs in monitoring intertidal zones. The major contributions are summarized as follows:

(1)  The impact of UAV flight pattern, altitude, and image overlap on the accuracy of intertidal topographic observations without ground control points was quantitatively assessed. This provides scientific guidelines for the balance between the accuracy and efficiency of UAV-based intertidal monitoring;

(2)  The errors caused by the water-bearing layer in low-lying mudflats were estimated and elevation corrections for the water-bearing areas were inferred from field measurements, thus ensuring the accuracy of topographic change monitoring in the mudflats;

(3)     Given the limited spatial scale of UAV mapping, the potential for combining UAV and satellite observations of mudflat topography was explored to take advantage of the high spatial and temporal accuracy of UAVs and the large spatial coverage of satellite sensor imagery.

## 2. Materials and Methods

### 2.1. Study Area

The Chongming Dongtan Nature Reserve (CDNR, 31.25°–31.38°N, 121.50°–122.05°E) is located at the eastern end of Chongming Island (Figure 1a), the largest estuarine alluvial island in the world. The estuarine wetlands at the CDNR, as the main habitat for migrant birds, are composed of mudflats in the low tidal zone and salt marshes in the middle and upper tidal zones. Existing human-induced coastal restoration projects promote sediment deposition and drive the progradation of shorelines. As a result, the intertidal topography in the CDNR was altered frequently and at short timescales [45]. A total of 20 transects with a width of 200 m have been designed for yearly topographic observations of mudflats since 2020 to investigate recent dynamics of erosion and accretion (Figure 1b). Prior to the topography monitoring, two experimental sites were selected to evaluate the impacts of flight parameters and water content variations on accuracy and underlying uncertainty. The first experimental site (site A; an area of 22 ha) is located in the north of the CDNR, where low-lying areas are twice inundated by tides daily, and thus a part of the mudflats retains a shallow water layer (about 2 cm) on the surface (Figure 1c). Another experimental site (site B, an area of 32 ha) with high elevation is located in the middle of the CDNR and is only completely inundated during the high tide over the spring tidal regime.

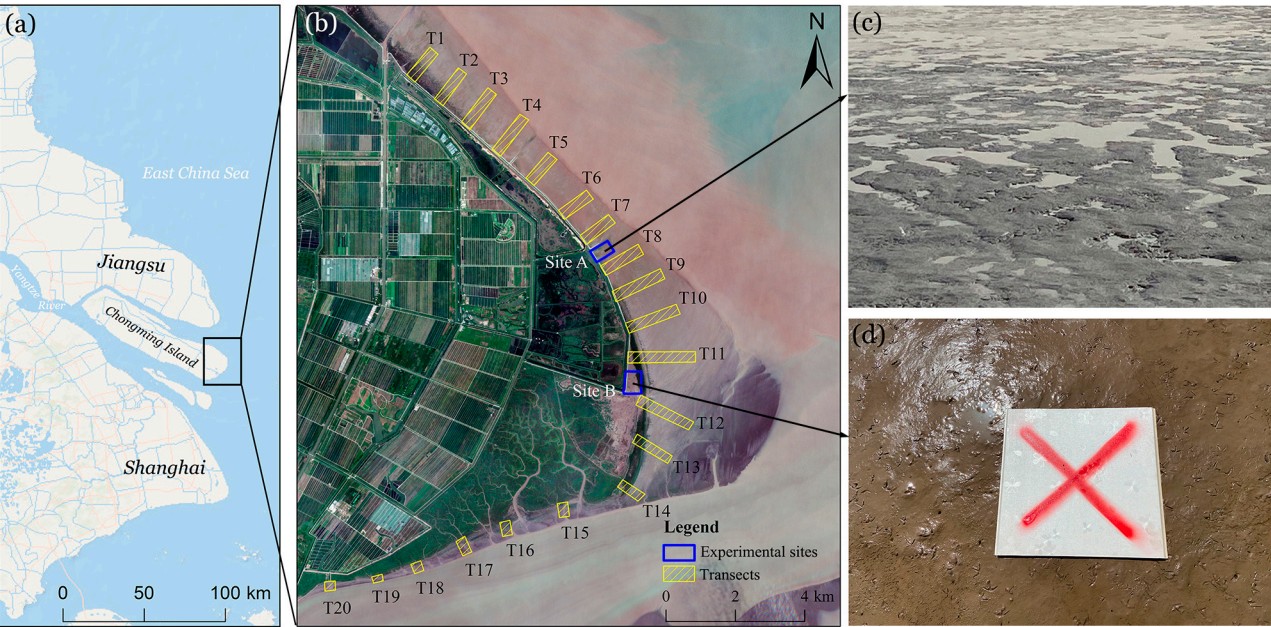

**Figure 1.** The study area in the Chongming Dongtan Nature Reserve, northern Chongming Island, China (**a**,**b**). The satellite image was acquired by Sentinel-2 on 4 July 2021; (**c**) shallow surface water layer on the low-lying mudflats at site A; and (**d**) no surface water accumulation of mudflats with higher elevation at site B.

### 2.2. Data and Methods

Previous studies have demonstrated that lower flight altitudes of UAVs and higher image overlap tend to reconstruct terrain with very high accuracy [19,31]. However, lower flight altitudes mean that a large amount of flight time is required, which poses a significant obstacle for intertidal mudflats with short exposure times and UAVs with low battery life. To explore how to well balance the efficiency and accuracy of UAV mapping, a series of

flight tests were designed prior to conducting the transect observations. Figure 2 illustrates the flowchart for quantifying mudflat topographic changes using an RTK-assisted UAV and an RTK mobile station. The terrain acquired by the UAV at a 50-m flight altitude with an 80% image overlap and vertical photogrammetry was used as a baseline. The terrains acquired by other photogrammetric configurations (different flight patterns, altitudes, and overlaps) were compared with the baseline, and then the optimal photogrammetric configuration that balanced accuracy and efficiency was selected for the annual topographic monitoring of CDNR. The histogram equalization and K-means clustering were used to automatically identify low-lying water-bearing areas, and error estimates and elevation corrections were performed based on field measurements. In addition, a UAV-based DEM profile and water levels from the tide gauge station were employed to assign elevations to time-series waterlines, respectively, in order to investigate the difference in the accuracy of the resulting DEMs.

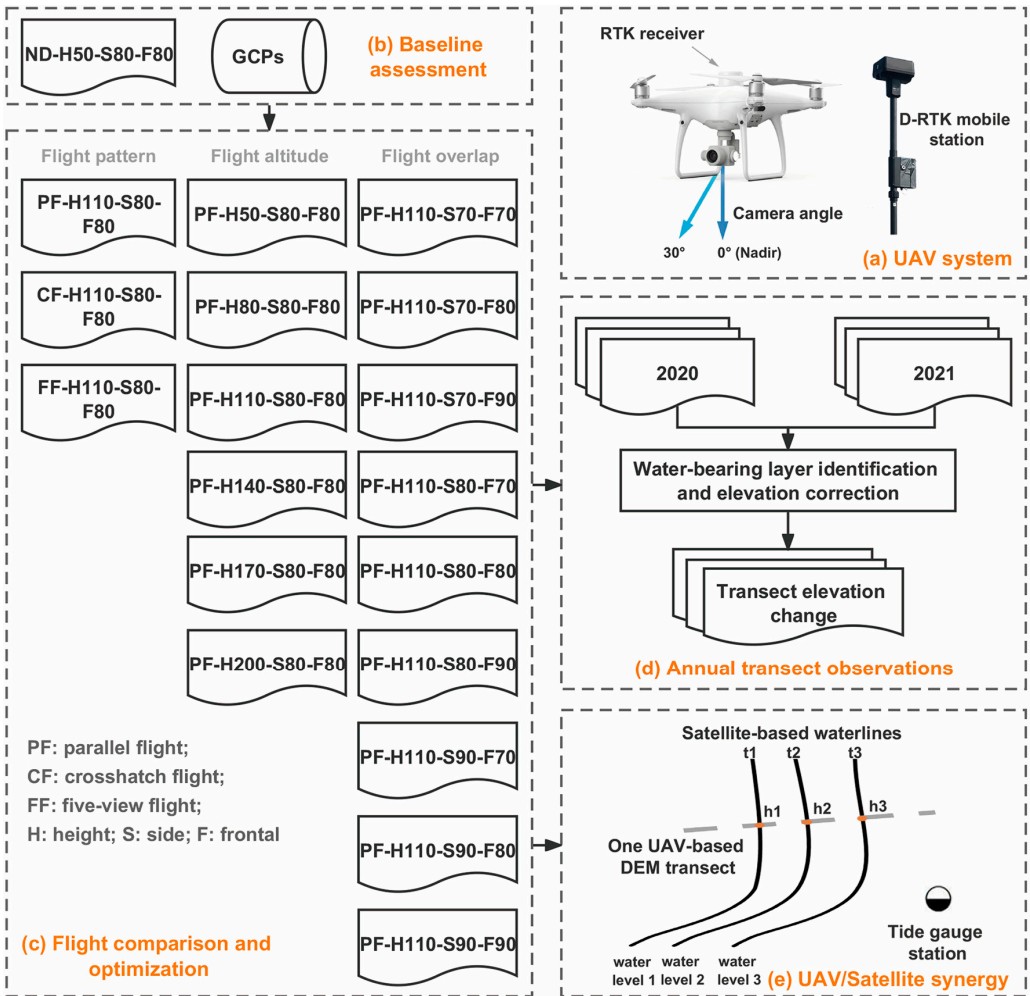

**Figure 2.** (**a**) The UAV system used in this study; (**b**–**d**) the workflow for accurate monitoring of mudflat topographic change; and (**e**) the schematic diagram of UAV/satellite synergy for mapping mudflat topography.

### 2.2.1. UAV Systems

The DJI Phantom 4 RTK Drone (P4R, Figure 2a) was used for image collection in this study. The P4R camera has a lens with a maximum focal length of 24 mm and a field of view (FOV) of 84°, which can capture high-resolution images with a 1-inch, 20-megapixel CMOS sensor. The lens has undergone a rigorous intrinsic calibration to measure radial and tangential lens distortions. This process was carried out by the camera manufacturer. The distortion parameters were automatically written into the image when the photograph

was taken. During the SfM process, the camera calibration parameters were used as initial values and were then further refined by aerial triangulation for accurate self-calibration of the camera. The RTK receiver on top of the drone can receive the real-time kinematic data forwarded from the remote controller via connection to the D-RTK 2 GNSS Mobile Base Station. This ensures 1 cm + 1 ppm RTK horizontal and 1.5 cm + 1 ppm RTK vertical positioning accuracy. In addition, the RTK positioning accuracy of the camera at the moment of UAV capture is also written to the image metadata and supplemented with the results of the photogrammetric outputs to verify the stability of the position and orientation system (POS). Such an accurate and stable POS for drones makes it possible to survey inaccessible mudflats without GCPs.

### 2.2.2. Photogrammetric Experiments and Image Processing

To assess the impacts of flight pattern, flight altitude, and image overlap of the flight path on elevation surveying of tidal flats, 16 flight experiments were performed at 2 study sites according to different photogrammetric configurations. Table 2 lists all the flights with their photogrammetric configurations, ground sample distance (GSD), and the number of photographs acquired. Three flight patterns (i.e., parallel flight, crosshatch flight, and five-view flight) were tested with other standardized flight settings (Figure 3). For the parallel flight, the UAV camera took photographs along parallel flight lines at a nadir angle ($0°$ or perpendicular to the ground; Figure 2a). The crosshatch flight means that the UAV took pictures along the crosshatch line at an off-nadir camera angle of $30°$, while the five-view flight consists of oblique photographs taken to the north, south, east, and west at an off-nadir camera angle of $30°$ and nadir photographs. In particular, for the parallel flight, the UAV took a set of oblique photographs to improve the calibration of the interior orientation elements by flying toward the center of the survey area at the end of the aerial survey (Figure 3a). Nine groups of control experimental flights at different frontal and side overlaps and six groups of control experimental flights at different flight altitudes were conducted. A total of 12 brightly colored 0.5 m × 0.5 m photogrammetric targets were positioned across site B before flying to serve as checkpoints and their coordinates were collected using the D-RTK 2 GNSS Mobile Base Station. In addition, one flight with the P4R drone was performed at site A to evaluate the influence of the water-bearing layer on the accuracy of photogrammetric results. After quantitative comparison and evaluation, an aerial survey method that balances accuracy and efficiency was selected to collect data at 20 fixed monitoring sections of CDNR to investigate the changing patterns of sedimentation from 2020 to 2021. All of the above flights were performed at low tide during spring tide and the ISO and shutter speed were set automatically. All UAV images were pre-processed using the SfM photogrammetry algorithm implemented by the Pix4Dmapper software, and all resultant datasets were re-projected to a common horizontal and vertical coordinate system (UTM 51N WGS84 with the EGM96 vertical datum).

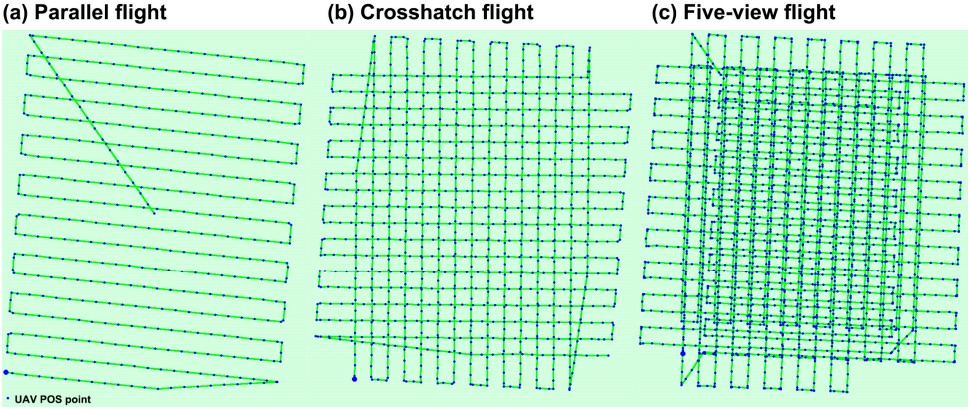

**Figure 3.** Flight pattern for (**a**) parallel flight, (**b**) crosshatch flight, and (**c**) five-view flight.

**Table 2.** UAV flight experiments with different photogrammetric configurations.

| Experiments | Flight Pattern | Altitude (m) | Side Overlap | Frontal Overlap | GSD (cm) | Number of Photographs |
|---|---|---|---|---|---|---|
| PF-H110-S80-F80 | Parallel | 110 | 80% | 80% | 3.1 | 444 |
| CF-H110-S80-F80 | Crosshatch | 110 | 80% | 80% | 3.8 | 1073 |
| FF-H110-S80-F80 | Five-view | 110 | 80% | 80% | 3.6 | 2551 |
| PF-H50-S80-F80 | Parallel | 50 | 80% | 80% | 1.3 | 2082 |
| PF-H80-S80-F80 | Parallel | 80 | 80% | 80% | 2.2 | 792 |
| PF-H140-S80-F80 | Parallel | 140 | 80% | 80% | 4.0 | 286 |
| PF-H170-S80-F80 | Parallel | 170 | 80% | 80% | 4.9 | 226 |
| PF-H200-S80-F80 | Parallel | 200 | 80% | 80% | 5.8 | 166 |
| PF-H110-S70-F70 | Parallel | 110 | 70% | 70% | 3.1 | 239 |
| PF-H110-S70-F80 | Parallel | 110 | 70% | 80% | 3.1 | 354 |
| PF-H110-S70-F90 | Parallel | 110 | 70% | 90% | 3.1 | 676 |
| PF-H110-S80-F70 | Parallel | 110 | 80% | 70% | 3.1 | 279 |
| PF-H110-S80-F90 | Parallel | 110 | 80% | 90% | 3.1 | 847 |
| PF-H110-S90-F70 | Parallel | 110 | 90% | 70% | 3.1 | 504 |
| PF-H110-S90-F80 | Parallel | 110 | 90% | 80% | 3.1 | 738 |
| PF-H110-S90-F90 | Parallel | 110 | 90% | 90% | 3.1 | 1407 |

### 2.2.3. Accuracy Assessment and Comparisons

SfM can resolve all camera positionings and scene geometry simultaneously using a highly redundant bundle adjustment based on matching features in multiple overlapping images. The goal of bundle adjustment is to minimize the reprojection error (RE) between predicted projections and their observed corresponding image points. The reprojection error output by the SfM algorithm under different photogrammetric configurations can, therefore, be used as a reference indicator for error assessment. The object function of bundle adjustment is presented as:

$$g(C, X) = \sum_{i=1}^{n} \sum_{j=1}^{m} \omega_{i,j} \parallel q_{i,j} - P(C_i, X_j) \parallel^2 \tag{1}$$

where $g(C, X)$ is the projection of point $X_j$ on the camera $C_i$; $q_{i,j}$ is an observed image point; $\omega_{i,j}$ is an indicator function with $\omega_{i,j} = 1$ if point $X_j$ is visible in the camera $C_i$; otherwise, $\omega_{i,j} = 0$. The root-mean-square error (RMSE) between checkpoints and the UAV-based DEM derived from a UAV flight at an altitude of 50 m was also calculated. This is especially true considering that a few validation points cannot fully reveal the accuracy difference in terrain obtained under different photogrammetric configurations. Therefore, the UAV-based DEM acquired at 50 m altitude was also used as a baseline and the RMSEs between it and the DEMs acquired at other flight conditions were calculated, respectively:

$$\text{RMSE} = \sqrt{\frac{1}{n} \sum_{i=1}^{n} (h_i - h_{i,\,50})^2} \tag{2}$$

where $h_i$ refers to the elevation value obtained from other UAV-based DEMs, $h_{i,\,50}$ represents the elevation measured from UAV-based DEM acquired at 50 m altitude, and n is the number of sample points. In addition, James et al. [46] indicated that the spatial variability of error should be assessed when using the RMSE, the standard deviation of the error (SDE) was, therefore, also calculated for measuring precision:

$$\text{SDE} = \sqrt{\frac{1}{n} \sum_{i=1}^{n} (e_i - \bar{e})^2} \tag{3}$$

where $e_i$ refers to the elevation error observed from other UAV-based DEMs and measured from UAV-based DEM acquired at 50 m altitude and $\bar{e}$ refers to the average of the errors. In

order to achieve the above accuracy comparison, experimental site B first meshed with a grid size of 1 × 1 m and then the elevation values at the center of the grid were extracted for point-to-point comparison within different DEMs with a total of 108,005 points. Prior to gridding, high-precision spatial alignment of the different DEM values should be performed to avoid elevation comparisons of non-synonymous points due to horizontal displacement. Firstly, the orthomosaic from the 50-m flight was used as a reference image, and then the corresponding feature points in the orthomosaics from other flight configurations were identified. Crab burrows throughout the mudflats were considered excellent alignment control points since they were adequately visible in orthomosaics with different spatial resolutions. A total of 17 corresponding points were found, evenly distributed in the orthomosaics. The corresponding DEM registration was then performed using a third-order polynomial model so that the residuals in the horizontal positions were less than 2 cm. This process was carried out using the georeferencing tool in ArcGIS.

### 2.2.4. Identification of Surface Water Layer

K-means clustering is a simple and versatile algorithm that can be used for image segmentation tasks. It can group similar pixels together without any prior knowledge. In this study, the water body pixels and non-water body pixels in the UAV orthomosaics show color differences, and the water body pixels have similar colors. Therefore, K-means clustering is well-suited for clustering these water body pixels together and can be partitioned into an output class. To accurately identify low-lying water-bearing areas in the UAV orthomosaics, histogram equalization was first used to enhance the contrast between water and non-water bodies, and then K-means clustering was utilized to classify the water bodies (Figure 4). The classification results were converted to vectors and small isolated patches were removed using ArcGIS. A total of 100 random points were generated from these classified surface pixels and their elevations were obtained from the DEM reconstructed by the images acquired simultaneously by the P4R UAV. The elevation of the adjacent water-free pixel for each random point was also measured manually and the difference in elevation between the pair was compared.

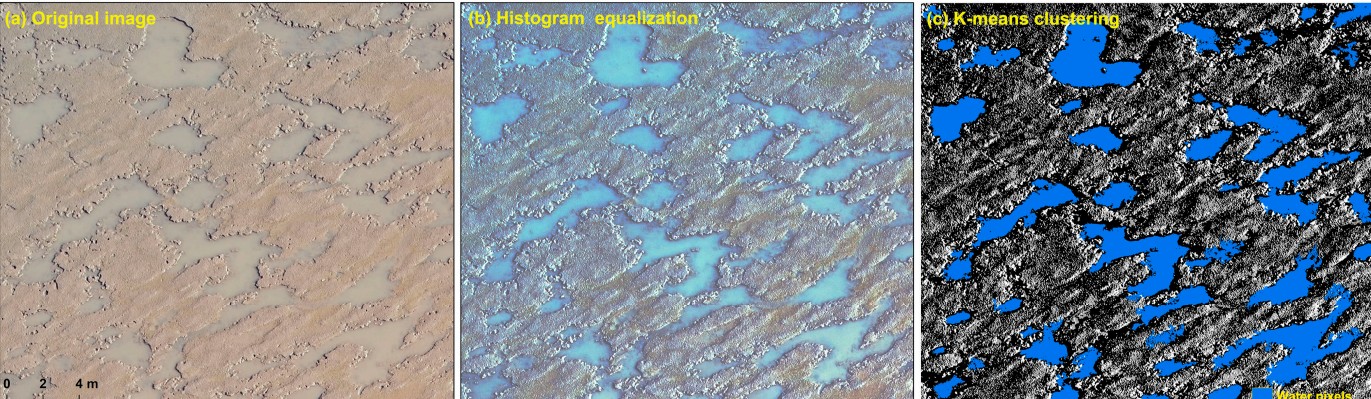

**Figure 4.** Identification of surface water areas using histogram equalization and K-means clustering.

### 2.2.5. Elevation Determination of Satellite-Based Waterlines from Photogrammetric DEMs

Although UAV photogrammetry can achieve high-precision and high-resolution topography for intertidal mudflats and track associated changes, it is limited in scale for mapping vast coastal mudflats. Commonly used satellite-based waterline methods have limited accuracy and spatiotemporal resolution. Here, we manually digitized 19 waterlines from time-series Sentinel-2 images between April 2020 and April 2021 to assess the integration of UAV and satellite Earth observation for monitoring intertidal mudflat topography. As shown in Figure 2e, elevation values of waterlines were determined using tide level data from Sheshan station and a UAV-based SfM photogrammetric elevation for sections observed in 2021 (i.e., T8 transect), respectively. Triangulated irregular networks (TINs)

were then constructed using the ArcGIS software. Next, the TINs were converted to raster to produce two DEMs for the whole of the mudflats in the Chongming Dongtan, including the tide-adjusted DEM and the UAV-adjusted DEM, respectively. A total of 44 corresponding points were extracted from these 2 DEMs to compare with other UAV-based SfM photogrammetric elevation sections observed in 2021 for accuracy comparison.

## 3. Results

### 3.1. Comparison of Different Photogrammetric Results

The horizontal locations and elevations of 12 checkpoints extracted from the UAV-based photogrammetric DEM acquired at a 50-m altitude were compared to RTK-GNSS measurements (Figure 5a). The horizontal displacement ranged from 1.0 to 4.2 cm with an X-directional RMSE of 3 cm and a Y-directional RMSE of 2.8 cm, while the elevation difference ranged from 1.6 to 5.8 cm with an RMSE of 3.1 cm. Table 3 provides the reprojection errors for the different photogrammetric configurations. We found that reprojection errors from aerial surveys at different flight altitudes and overlap levels hardly differ significantly at the pixel scale, but there are differences in reprojection errors at the spatial scale for aerial surveys at different flight altitudes because the flight altitude determines the ground sampling distance (i.e., UAV image resolution). However, differences in flight pattern, altitude, and side and frontal overlaps all have a significant impact on the accuracy of the final DEMs generated from UAV images. It is generally agreed that five-view flight photogrammetry provides the most surface information and a high level of terrain reconstruction accuracy. However, our flight experiments showed that for the same flight altitude and image overlap conditions, the highest accuracy of UAV-based terrain reconstruction was achieved by the parallel flight, and the crosshatch photogrammetry produced the lowest accuracy for terrain reconstruction. Comparisons of DEMs constructed by UAV images acquired at different flight altitudes show that the higher the flight altitude, the lower the accuracy of the constructed DEM, but the accuracy is non-linear in relation to changes in altitude. Compared to the DEM reconstructed from UAV images at 50 m flight altitude, the DEMs accuracy generated at 80 m, 110 m, and 140 m flight altitude varied little, with both RMSEs and SDEs between 2 and 3 cm. When the flight altitude reached 170 m and 200 m, the accuracy of the generated DEMs lost a considerable amount of accuracy, with RMSEs of 3.5 cm and 5.9 cm, and SDEs of 2.5 cm and 5.8 cm, respectively. In terms of image overlap, the increase from 70% to 80% in frontal overlap and side overlap resulted in a significant increase in accuracy of 1–2 cm. However, the accuracy varies very little from 80% to 90%. The quantitative evaluation of the above flight experiments showed that the selection of the appropriate aerial photogrammetric mode, flight altitude, and image overlap can save time in data collection while maintaining a high level of accuracy, which is important for observations in the very short exposure time of the intertidal zone. As seen in Figure 5b, the UAV could balance accuracy and efficiency well with an 80% image overlap and off-nadir photogrammetry at a flight altitude of 110 m.

**Table 3.** Reprojection error, the root-mean-square error, and the standard deviation of error for each of UAV flight experiments with different photogrammetric configurations.

| Experiments | RE (pixel/cm) | RMSE (cm) | SDE (cm) | Experiments | RE (pixel/cm) | RMSE (cm) | SDE (cm) |
|---|---|---|---|---|---|---|---|
| PF-H110-S80-F80 | 0.134/0.415 | 2.5 | 2.4 | PF-H110-S70-F70 | 0.136/0.422 | 5.4 | 1.8 |
| CF-H110-S80-F80 | 0.090/0.342 | 5.4 | 2.6 | PF-H110-S70-F80 | 0.137/0.425 | 4.5 | 2.6 |
| FF-H110-S80-F80 | 0.096/0.346 | 3.6 | 2.4 | PF-H110-S70-F90 | 0.099/0.307 | 4.4 | 2.5 |
| PF-H50-S80-F80 | 0.097/0.126 | / | / | PF-H110-S80-F70 | 0.144/0.446 | 5.0 | 2.2 |
| PF-H80-S80-F80 | 0.101/0.224 | 2.1 | 2.1 | PF-H110-S80-F90 | 0.104/0.322 | 2.3 | 1.8 |
| PF-H140-S80-F80 | 0.142/0.568 | 2.7 | 2.5 | PF-H110-S90-F70 | 0.111/0.344 | 3.8 | 3.7 |
| PF-H170-S80-F80 | 0.141/0.691 | 3.5 | 2.5 | PF-H110-S90-F80 | 0.108/0.335 | 2.2 | 1.6 |
| PF-H200-S80-F80 | 0.137/0.795 | 5.9 | 5.8 | PF-H110-S90-F90 | 0.103/0.319 | 2.1 | 2.1 |

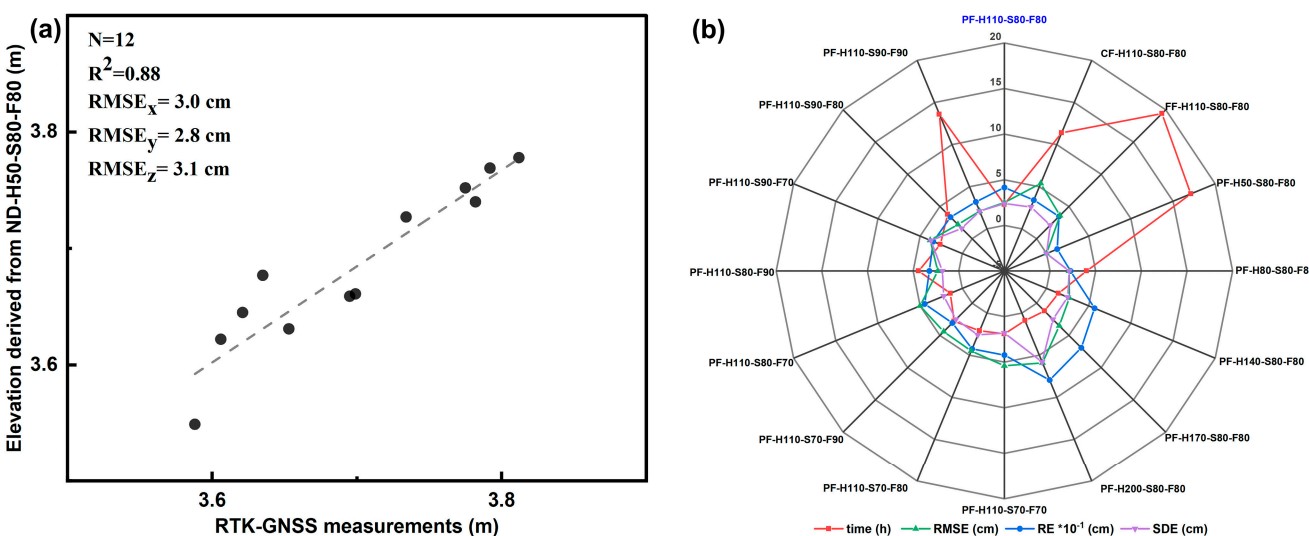

**Figure 5.** (**a**) Comparison between the elevations of checkpoints extracted from UAV-based photogrammetric DEM acquired at 50 m altitude and RTK-GNSS measurements; and (**b**) UAV photogrammetric accuracy and efficiency under different flights.

### 3.2. The Impact of Surface Water Layer on UAV-Based DEM

In general, areas of the water body areas are missing from photogrammetric reconstructions, and even if they are not, the reconstructions are anomalous in the vertical direction. However, in our experiments, we found that the shallow, turbid, and small extent of the water body did not significantly hinder the detection and matching of image key points (Figure 6a). As a result, no significant elevation outliers were found in the water-bearing regions of the generated DEMs. Furthermore, although the DEMs produced by UAV photogrammetry were the result of smoothing after attempts at pixel-wise image matching, they could still represent well the topographic features of the mudflats and their magnitude of variations well, such as the formation of small tidal creeks and the traces of siltation or scour on the mudflat surface, thanks to their centimeter-level spatial resolution (Figure 6b–d).

When overlaying the identified water pixel boundaries with the orthophotos and DEMs obtained by P4R revealed that the elevation of the area where the water pixels are located is lower (Figure 7b), which is consistent with reality. It was also found that the water surface elevation values in the same area varied within approximately 2 cm, and no significant elevation anomalies were found due to the presence of the water surface. By comparing the elevation values of water pixels with their adjacent non-water pixels, it was found that 99% of the non-water pixels had an elevation greater than that of their adjacent water pixels, with an average bias of 4.6 cm (Figure 7c). In addition, field measurements of the height of the water surface from the non-water surface and the thickness of the water-bearing layer along the water boundary revealed that the height of the water surface from the non-water surface was approximately 5 cm and the thickness of the water layer was approximately 2 cm (Figure 7d). This suggested that the elevation generated by UAV photogrammetry were overestimated in the low-lying water-bearing regions, with an overestimate of approximately 2.4 cm. Therefore, a statistical-based elevation correction was applied to the water-bearing areas during the processing of the transect DEMs to accurately detect elevation changes.

### 3.3. The Pattern of Accretion/Erosion in Chongming Dongtan

Based on the above quantitative assessment results, the UAV flights with nadir photogrammetry, a flight altitude of 110 m, and 80% image overlap were conducted at the fixed observation section of the CDNR in May 2020 and 2021. Figure 8 shows the spatially distinct patterns of accretion and erosion in the mudflats of the CDNR and the average

annual change in elevation for each transect. From 2020 to 2021, the mudflat topography alternated between siltation and erosion from the north to the south of the CDNR. The most severe erosion occurred in the southern part of Dongtan with an average annual erosion rate of over 0.1 m/year. The most rapid sedimentation occurred in the central part of Dongtan, with an average annual sedimentation rate of 0.12 m/year. For the northern eroded transects, erosion was more severe on the landward side with an average annual erosion rate of over 0.1 m/year (e.g., T7 and T8). The central transects have experienced sedimentation (T9–T14) and sedimentation was greatest on the seaward side with an average annual siltation rate of over 0.2 m/year (e.g., T11 and T12). In contrast, the southern transects were all subject to erosion with an average annual erosion rate of 0.1–0.2 m/year. In addition, we found that the erosion or sedimentation of a section was consistent with the width of the tidal flats. This means that the eroded transects were located on tidal flats with a relatively small seaward width, while the transects on the tidal flats with a larger seaward width were undergoing rapid sedimentation.

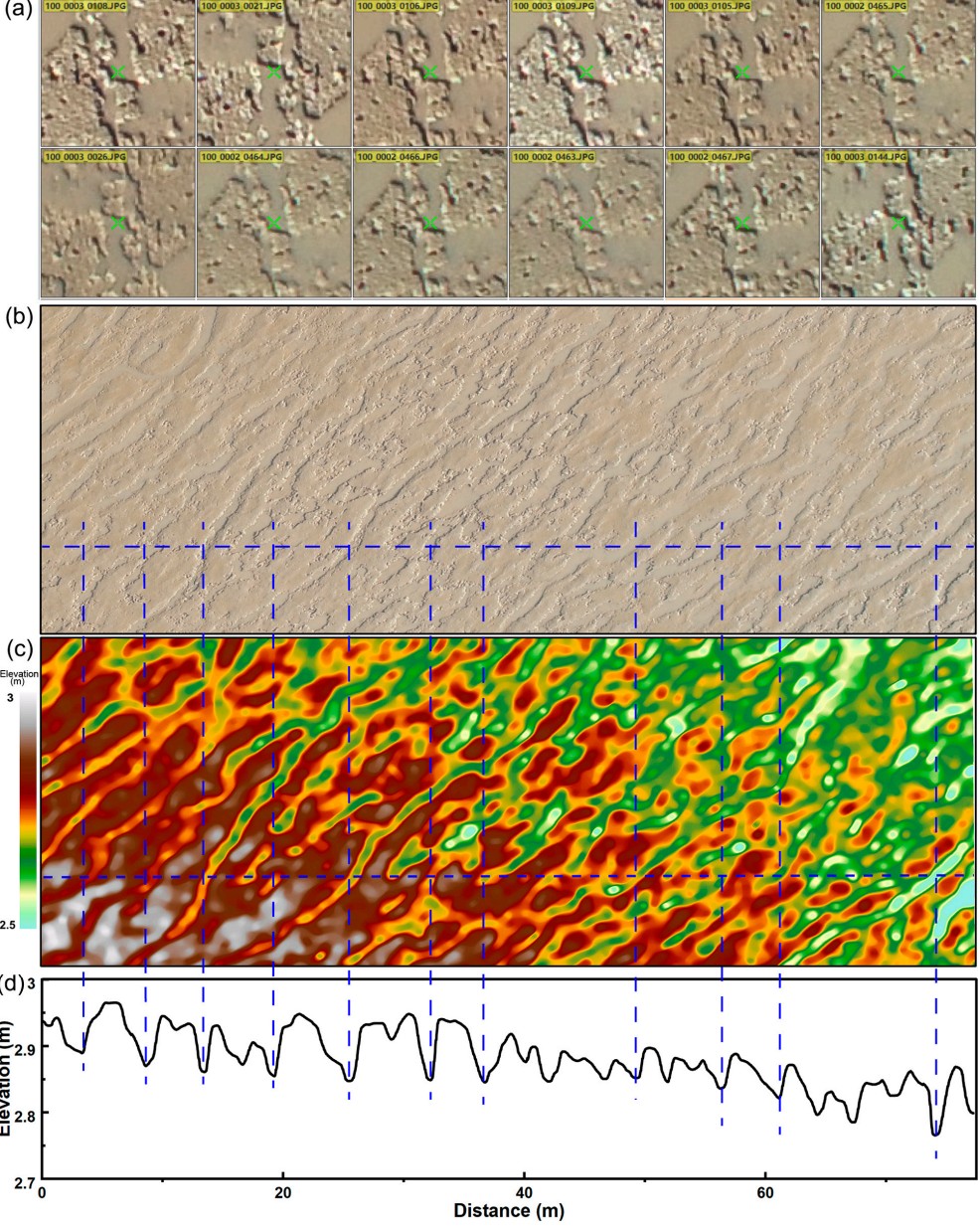

**Figure 6.** (**a**) The detection and matching of image key points in water-bearing areas; (**b**) the UAV-based orthomosaic; (**c**) the UAV-based DEM; and (**d**) elevations along one profile.

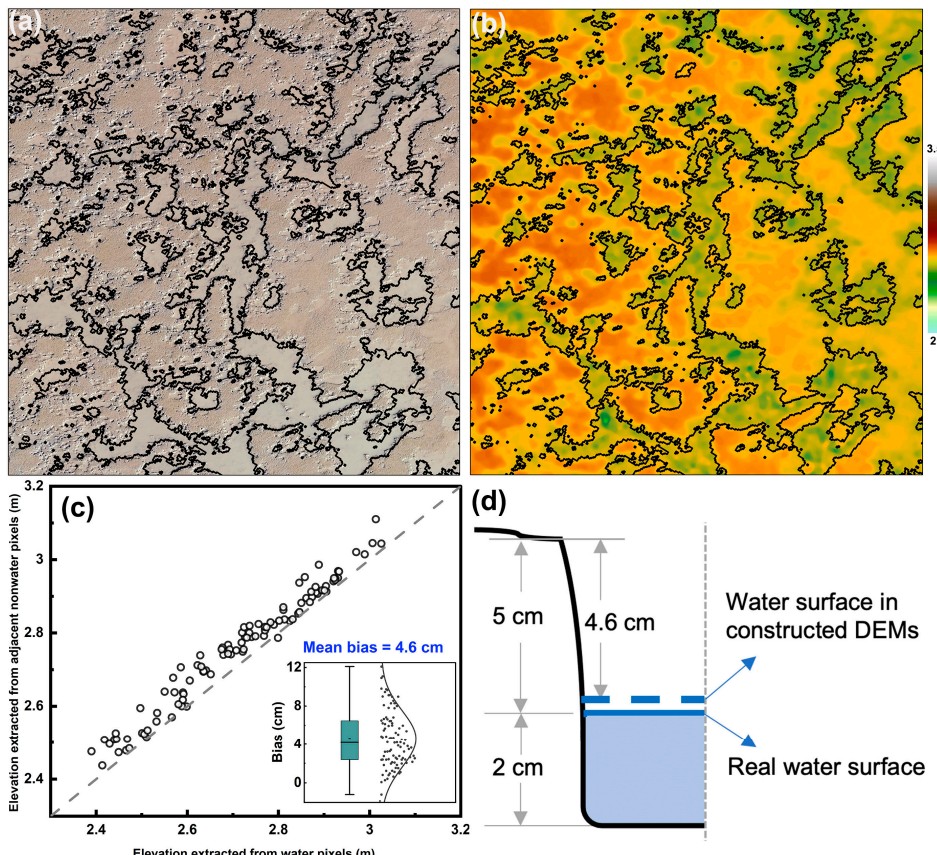

**Figure 7.** The impact of surface water-bearing layer on UAV-based DEM: (**a**) Water pixel identification results overlayed on the orthophotos; (**b**) water pixel identification results overlayed on UAV-based DEM; (**c**) comparison between elevations of water pixels and adjacent non-water pixels; and (**d**) the semi-profile schematic of ground elevation overestimation in water areas.

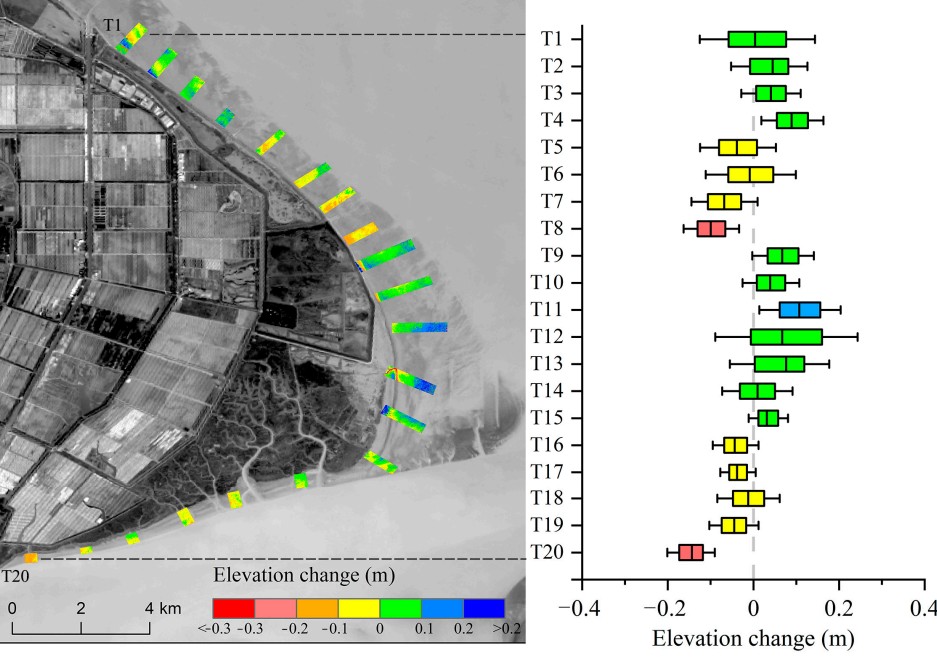

**Figure 8.** The patterns of accretion and erosion in the CDNR and the average annual elevation change of transects from May 2020 to May 2021.

### 3.4. Accuracy Comparison of Tide-Adjusted DEM and the UAV-Adjusted DEM

Figure 9a shows the multi-temporal waterlines extracted from the time-series Sentinel-2 imagery. The waterlines were assigned elevation values using water level data from the tide gauge station or a section of elevation obtained by the UAV photogrammetry and then interpolated to create the tide-adjusted mudflat DEM (Figure 9b) and the UAV-adjusted mudflat DEM (Figure 9d). A comparison of these two DEMs with the other UAV photogrammetric section elevations indicates that the RMSEs are 47 cm and 23 cm for the tide-adjusted DEM and the UAV-adjusted DEM, respectively (Figure 9c). This means that the UAV photogrammetric results for the determination of waterline elevations can increase the accuracy of traditional waterline methods by up to 51% significantly. Thus, UAV/satellite synergy can not only make use of the flexibility for drone surveys, but also achieve high accuracy for large-scale intertidal topography monitoring.

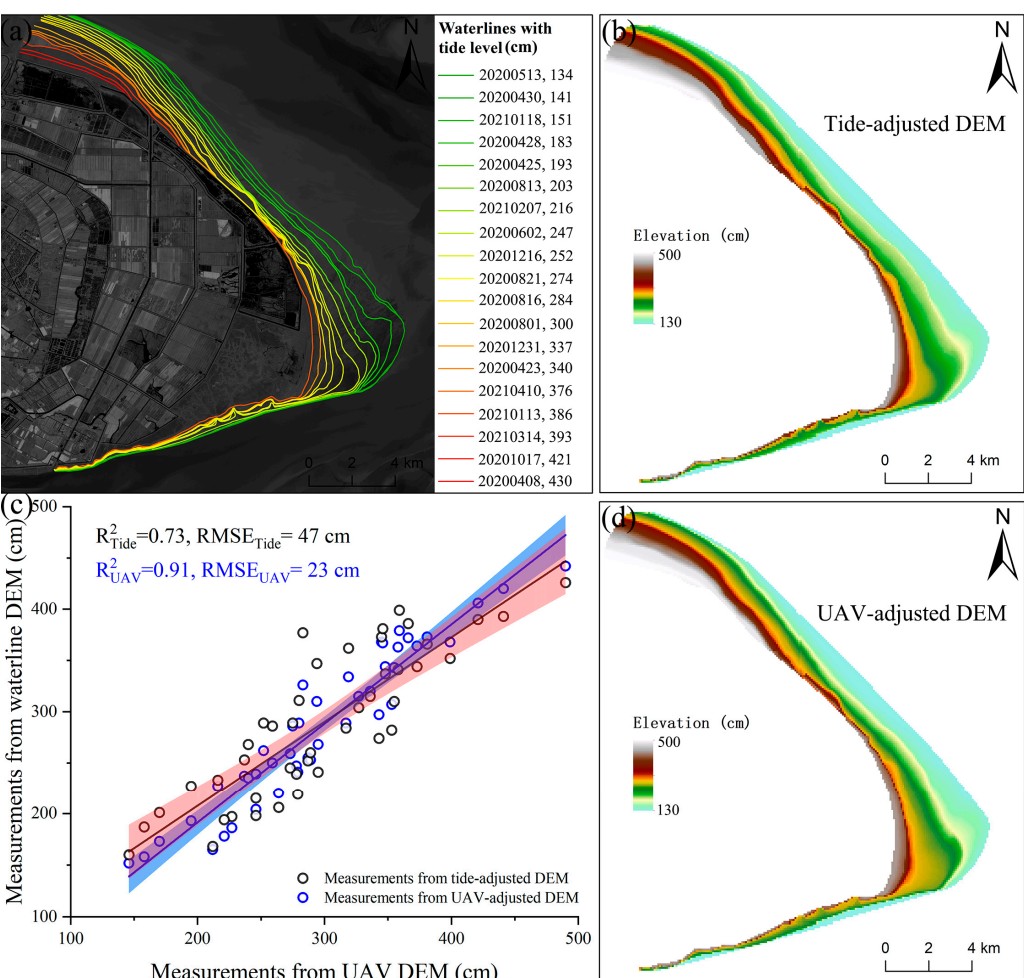

**Figure 9.** Mudflat DEM derived from the satellite-based waterline method: (**a**) A series of waterlines digitized from Sentinel-2 images; (**b**) DEM generation using tide levels to assign waterline elevations; (**c**) comparison of DEM accuracy produced by different waterline elevation assignments; and (**d**) DEM generation using UAV-based SfM profile to assign waterline elevations.

## 4. Discussion

### 4.1. Uncertainty Caused by Photogrammetric Configurations

The main uncertainty in UAV-based photogrammetry without using GCPs for monitoring intertidal topography arises from the accuracy in resolving the interior and exterior orientation elements of the UAV camera [47,48]. The camera of the P4R UAV used in this study has been rigorously calibrated and the outputs of all flight experiments processed by the Pix4Dmapper software show that there is no significant difference in the internal



orientation elements solved by the overall bundle adjustment. The accuracy of the exterior orientation elements depends primarily on the positioning accuracy at the moment when the UAV takes a picture, where the time synchronization between the UAV positioning module and the camera module is particularly important, as a millisecond of time synchronization error will result in a position displacement of several centimeters at a flight speed of tens of meters per second. In this study, the UAV was connected to the D-RTK 2 GNSS Mobile Base Station for real-time positioning with an accuracy of better than 2 cm, thus allowing it to be used for direct positioning in the context of UAV photogrammetry.

In the case of consistent uncertainty of the UAV orientation elements, the uncertainty caused by the UAV flight pattern, altitude, and image overlap can potentially influence the critical first step of the SfM algorithm, i.e., key point extraction and matching. Figure 10 shows the average number of extracted key points per image and the number of successful matchings for the different flight patterns. The average number of key points extracted per image is comparable for the three flight patterns, but the average number of successfully matched key points per image varies considerably. The number of successfully matched key points per image obtained by parallel flight with a nadir camera angle is twice that of the other flight patterns, possibly indicating its high reconstruction accuracy at other non-feature points obtained by smoothing in dense matching. For oblique UAV images, perspective deformations may introduce more false key point matches, especially in tidal flats with similar repetition of textures [49]. The statistics also show that the success rate for key point matching was 23% for oblique images obtained by crosshatch flight, whereas 46% for nadir images obtained by parallel flight. As a result, vertical nadir photogrammetry achieved the best accuracy in three flight tests and the accuracy was significantly reduced with the addition of a large number of oblique images (i.e., the five-view flight photogrammetry). However, in some scenarios with large elevation fluctuations (e.g., mountainous regions and built-up areas), oblique images are necessary because information from different angles of the feature is required to reconstruct its full three-dimensional extent [50,51].

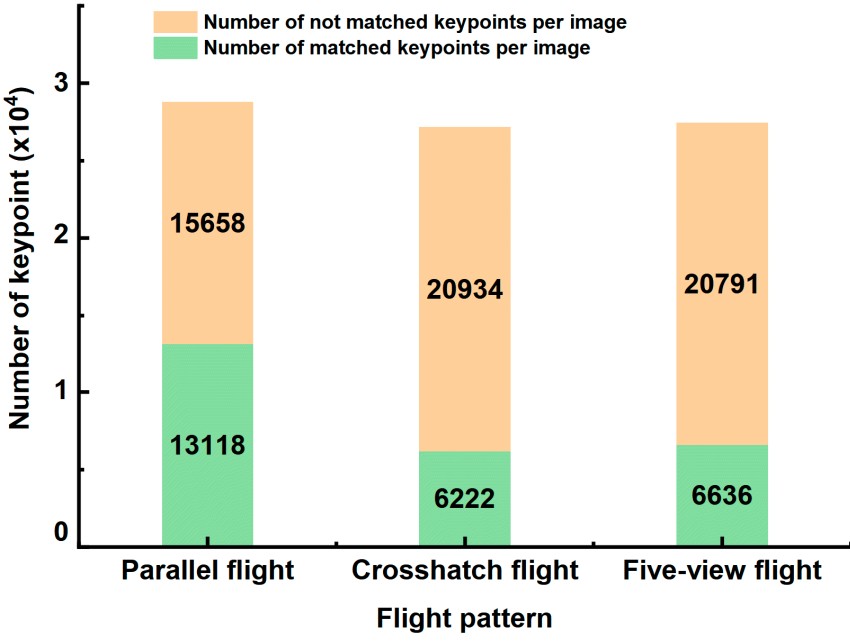

**Figure 10.** The average number of detected key points and matched points per image acquired under different flight patterns.

Flight altitude can affect the resulting topographic accuracy significantly, as flight altitude determines the image resolution. The key points used for the SfM are commonly derived from key point extraction algorithms (e.g., the Scale Invariant Feature Transform

(SIFT)), with sub-pixel positioning accuracy [52]. However, although the accuracy of key point extraction is comparable at the pixel scale, it varies greatly at the spatial scale due to the difference in detection scale. As a result, the absolute spatial accuracy of the key point detection is higher on the images acquired by the UAV at low altitudes, and the error propagated to the resulting DEM in the SfM process is smaller. For image overlap, higher overlap degrees (e.g., 90%) in both the side and frontal directions achieved better topographic accuracy because they ensure a stable image connection for the aerial triangulation. However, lower flight altitudes and higher image overlap will impose more time costs. In this study, we found that a UAV flight altitude of 110 m with 80% image overlap can provide a good compromise between accuracy and efficiency, and the resolution of the generated DEMs is sufficient to characterize the intertidal microtopography (e.g., small tidal channel) (Figure 11).

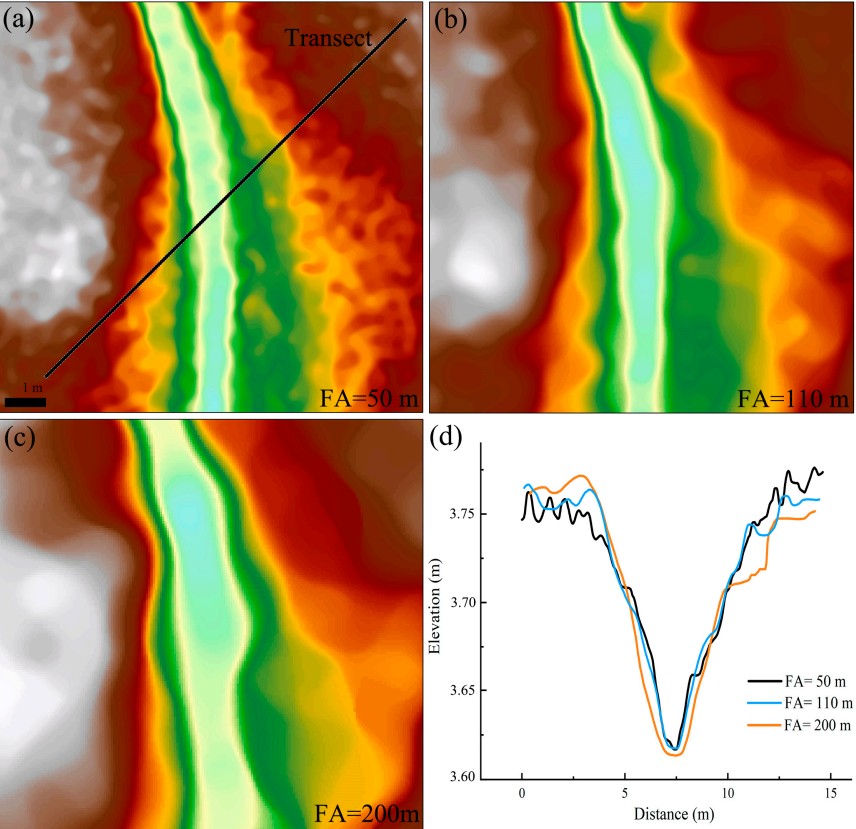

**Figure 11.** The DEMs generated by different UAV flight altitudes: (**a**) 50 m; (**b**) 110 m; (**c**) 200 m; and (**d**) small tidal channel profiles derived from DEMs.

### 4.2. The Potential and Challenges of UAV/Satellite Synergy

The potential errors of satellite-based waterline methods for reconstructing the large-scale mudflat topography arise mainly from waterline delineation, determination of waterline elevations, and waterline interpolation [53]. For mudflats with a slope of 1/1000, the horizontal displacement of one Sentinel-2 pixel (i.e., 10 m) caused by waterline extraction would result in an error of 1 cm in elevation. Errors of this magnitude can be tolerated provided that the accuracy of the waterline extraction is maintained. The errors introduced by the waterline interpolation can be mitigated by increasing the number of waterlines [54–56]. However, the errors introduced by the use of in situ measured or simulated tidal levels to assign elevation values to the waterlines cannot be ignored. Most of the tide stations are located some distance from the mudflats and therefore the tide station data does not truly match the tidal level at the moment of acquisition of the waterline. For areas without tide stations, hydrodynamic models are usually used to simulate tide levels, however, the lack of accurate nearshore bottom topography leads to simulation errors of up to tens of centime-

ters, especially in shallow nearshore areas with complex bathymetry and geometry [57,58]. Fortunately, UAV-based SfM photogrammetry can obtain the elevation for sections quickly and flexibly. Such elevation profiles can, therefore, be used to calibrate the elevation of waterlines, and based on our comparison, this approach can significantly increase mapping accuracy. Therefore, UAV/satellite synergy can not only bring out the high precision and flexibility of drones but also perform large-scale intertidal topography monitoring with considerable accuracy. The joint multi-source satellite data (e.g., Landsat 7/8/9, Sentinel-1/2) can obtain dense waterlines in a short period, supplemented by a cross-sectional elevation collected by a UAV during that period, which will greatly increase the temporal resolution and efficiency of intertidal topography monitoring by the UAV/satellite synergy. However, there are challenges in terms of how to quickly and automatically extract waterlines from satellite images from different sensors and keep their resultant uncertainty consistent. As shown in the study by de Vries et al. [59], accurately delineating instantaneous waterlines on estuarine coasts with very high sediment concentrations remains challenging due to the spectral similarity and radar backscatter complexity between subtidal- and intertidal features, and should be given more attention in future studies. Once the above problems are solved, UAVs can be used to obtain high-precision topography at the transect scale in the future, and then combined with dense satellite observations can detect hotspots and general trends of intertidal topographic changes in response to sea level rise and riverine sediment supply variations.

## 5. Conclusions

This study has quantitatively assessed the ability of RTK-assisted UAVs for surveying tidal flat topography without the use of ground control points; and the effects of flight pattern, altitude, and image overlap on topographic accuracy. These results demonstrate the availability of the RTK-assisted UAV removes the need for any additional ground control points. Unexpectedly, the parallel flight gives the highest accuracy and the crosshatch flight significantly reduces the accuracy. Lower flight altitudes and higher image overlap could improve the accuracy of UAV-based DEMs, while it will impose further time costs. Surface water in low-lying areas of intertidal mudflats does not cause anomalies in the three-dimensional reconstruction using the SfM algorithm, but it can result in an overestimation of elevation by several centimeters. In summary, the accuracy of UAV surveying in the intertidal zone is comparable to that of LiDAR, and the accuracy is controllable under uncertainty. In addition, the combination of UAV and satellite observation can construct high-precision and large-scale intertidal topography with significantly improved accuracy. The synergy of UAV and satellite applications can play a major role in identifying coastal erosion hotspots, establishing priority protection mechanisms, and facilitating coastal restoration in future work.

**Author Contributions:** Conceptualization, C.C. and C.Z.; methodology, C.C.; software, Y.D.; validation, C.C. and B.T.; formal analysis, C.C.; investigation, C.C. and W.W.; resources, B.T.; data curation, C.Z.; writing—original draft preparation, C.C.; writing—review and editing, C.C., C.Z. and W.W.; visualization, C.C.; supervision, B.T. and C.Z.; project administration, Y.Z.; funding acquisition, C.Z. and B.T. All authors have read and agreed to the published version of the manuscript.

**Funding:** This work was supported by the Natural Environment Research Council [grant number NE/T004002/1]. The research was partially funded by the project "Coping with Deltas in Transition" within the Programme of Strategic Scientific Alliances between China and the Netherlands (PSA), financed by the Ministry of Science and Technology of the People's Republic of China (MOST) [grant number 2016YFE0133700], and also sponsored by the China Scholarship Council (CSC).

**Data Availability Statement:** Not applicable.

**Conflicts of Interest:** The authors declare no conflict of interest.

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
