# Peer review of "UAV Photogrammetry in Intertidal Mudflats: Accuracy, Efficiency, and Potential for Integration with Satellite Imagery"

_remotesensing, doi:10.3390/rs15071814_

Round 1

Reviewer 1 Report

All my comments and suggestions are highlighted in the attached pdf

Author Response

Thank you for your comments on our manuscript entitled “UAV Photogrammetry in Intertidal Mudflats: Accuracy, Efficiency and Potential for Integration with Satellites” (ID: 2271957). Those comments are very helpful for revising and improving our paper. We have studied the comments carefully and made corrections which we hope meet with approval. Please find details in our attached response letter.

Reviewer 2 Report

UAV Photogrammetry in Intertidal Mudflats: Accuracy, Efficiency and Potential for Integration with Satellites

The inserted title of the manuscript does not totally match the included text and analysis because, the manuscript focused in Topographic Mapping of Intertidal Coastal Zone Using UAV is selected area …..

The methodology, which used in this manuscript, is unclear and it is not applicable.

English editing is need “some sentences is difficult to understand like are becoming increasingly viable economically and ecologically for reducing risk from coastal hazards. (Page3-line 49)

Page 6 Line 222-225: The authors said aerial survey method that balances accuracy and efficiency was selected to collect data at 20 fixed monitoring sections of CDNR to investigate the changing patterns of sedimentation from 2020 to 2021. (There is no sedimentation changing Pattern in this manuscript)

Page 9 line 222: the author indicate that time-series Sentinel-2 images between April 2020 and April 2021 used to assess the integration of UAV and satellite Earth observation for monitoring intertidal mudflat topography. However, UAV flights was in May 2020 and 2021. This monthly difference in data collection time could give unreliable results especially in tidal topography study.

Figure 6a: is not clear.

Figure 7: author must mention the techniques used to segment the water feature from DEM.

Figure 8. Need to more clarification, the topography along transects need to more analysis.

In figure 9: the source of Tidal data is not clear, what is the methodology applied for shoreline extraction from sentinel imagery and how shorelines combined with tidal range in the same period. Need to clarify.

In page 15 line 413 : The RMSe values is too large according to the accuracy of UAV images, It is recommended to use the same period for both UAV Flights and Sentinel imagery analysis.

It is recommended to calibrate and assess the produced UAV-based DEM should be calibrated and accuracy assessed with LiDAR or ground surveying.

The technique used to building DEM from the tidal data need to more references to indicate its effective way.

Author Response

(The authors gave the same response as above.)

Reviewer 3 Report

The authors submitted a well written and an interesting manuscript dealing with Accuracy, Efficiency and Potential UAV Photogrammetry data acquired in Intertidal Mudflats. The methodology is well described, and the conclusions are supported by the results. Below are some comments and suggestions to improve the overall quality of the manuscript.

Lines 1-2: on the title of the manuscript, “UAV Photogrammetry in Intertidal Mudflats: Accuracy, 2 Efficiency and Potential for Integration with Satellites”, please specify if you mean “Satellite Images, or Satellite observations” and correct the title to: “UAV Photogrammetry in Intertidal Mudflats: Accuracy, 2 Efficiency and Potential for Integration with Satellite images”.

Lines 87-88: Please provide references of recent studies conducted dealing with SfM reconstructs the three-dimensional structure of a scene or object, because they are more works conducted in this field other than the reference [33] of the work published in 1979. Please refer to (1) Muzirafuti, A.; Randazzo, G.; Maria, C.; Lanza, S. UAV Photogrammetry-Based Mapping of the Pocket Beaches of Isola Bella Bay, Taormina (Eastern Sicily); The Institute of Electrical and Electronics Engineers (IEEE): Calabria, Italy, 2021; (2) Beselly, S.M.; van der Wegen, M.; Grueters, U.; Reyns, J.; Dijkstra, J.; Roelvink, D. Eleven Years of Mangrove–Mudflat Dynamics on the Mud Volcano-Induced Prograding Delta in East Java, Indonesia: Integrating UAV and Satellite Imagery. Remote Sens. 202113, 1084. https://doi.org/10.3390/rs13061084.

Line 143: It would be better if you can put the section of Study Area as the subsection in the Section of Material and methods and continue this section with subsections of  dataset description; and methodology.

Line 302: It would be better if you could not start this section of results with Figures and Tables, please relocate them.  

Author Response

(The authors gave the same response as above.)

Round 2

Reviewer 2 Report

The title of study should be change to UAV Photogrammetry for Intertidal Mudflats Topographic Changes in Chongming Island, China

Figure 1 (b) source of data and accusation date, Photos (c, and d) need to specific location and time

Page 4 line 174. It recommended adding details for “The terrains
acquired by other photogrammetric configurations and references “

Figure 2. Should be move to .2.1. UAV Systems section

Figure 4: color scale is required

Author Response

Thank you for your comments on our manuscript entitled “UAV Photogrammetry in Intertidal Mudflats: Accuracy, Efficiency and Potential for Integration with Satellite Imagery” (ID: 2271957). Those comments are very helpful for revising and improving our paper. We have studied the comments carefully and made corrections which we hope meet with approval. The amendments are marked in red in the revised manuscript.
